# Molecular Characterization of a Rare Case of Bilateral Vitreoretinal T Cell Lymphoma through Vitreous Liquid Biopsy

**DOI:** 10.3390/ijms22116099

**Published:** 2021-06-05

**Authors:** Andi K. Cani, Marcus A. Toral, Daniel A. Balikov, Bryan L. Betz, Kevin Hu, Chia-Jen Liu, Matthew V. Prifti, Arul M. Chinnaiyan, Scott A. Tomlins, Vinit B. Mahajan, Rajesh C. Rao

**Affiliations:** 1Department of Internal Medicine, University of Michigan, Ann Arbor, MI 48105, USA; acani@med.umich.edu; 2Rogel Cancer Center, University of Michigan, Ann Arbor, MI 48109, USA; arul@med.umich.edu; 3Medical Scientist Training Program, University of Iowa, Iowa City, IA 52242, USA; marcustoral7@gmail.com; 4Graduate Program in Molecular Medicine, University of Iowa, Iowa City, IA 52242, USA; 5Molecular Surgery Laboratory, Byers Eye Institute, Stanford University, Palo Alto, CA 94303, USA; 6W.K. Kellogg Eye Center, Department of Ophthalmology and Visual Science, University of Michigan, Ann Arbor, MI 48105, USA; dbalikov@med.umich.edu; 7Department of Pathology, University of Michigan, Ann Arbor, MI 48109, USA; bbetz@med.umich.edu (B.L.B.); kevhu@med.umich.edu (K.H.); jiuchiaj@med.umich.edu (C.-J.L.); scott@strataoncology.com (S.A.T.); 8Department of Computational Medicine and Bioinformatics, University of Michigan, Ann Arbor, MI 48109, USA; 9Michigan Center for Translational Pathology, University of Michigan, Ann Arbor, MI 48109, USA; 10A. Alfred Taubman Medical Research Institute, University of Michigan, Ann Arbor, MI 48105, USA; matthew.prifti@wayne.edu; 11Department of Biological Sciences, Wayne State University, Detroit, MI 48202, USA; 12Palo Alto Veterans Health Care System, Palo Alto, CA 94304, USA; 13Division of Ophthalmology, Surgical Service, Veterans Administration Ann Arbor Healthcare System, Ann Arbor, MI 48105, USA

**Keywords:** precision oncology, next-generation sequencing, liquid biopsy, vitreoretinal lymphoma, T cell lymphoma

## Abstract

Vitreoretinal lymphoma (VRL) is an uncommon eye malignancy, and VRLs of T cell origin are rare. They are difficult to treat, and their molecular underpinnings, including actionable genomic alterations, remain to be elucidated. At present, vitreous fluid liquid biopsies represent a valuable VRL sample for molecular analysis to study VRLs. In this study, we report the molecular diagnostic workup of a rare case of bilateral T cell VRL and characterize its genomic landscape, including identification of potentially targetable alterations. Using next-generation sequencing of vitreous-derived DNA with a pan-cancer 126-gene panel, we found a copy number gain of *BRAF* and copy number loss of tumor suppressor *DNMT3A*. To the best of our knowledge, this represents the first exploration of the T cell VRL cancer genome and supports vitreous liquid biopsy as a suitable approach for precision oncology treatments.

## 1. Introduction

Vitreoretinal lymphoma (VRL), the most common ocular lymphoma, is often associated with primary CNS lymphoma (PCNSL, up to 90%) from where it is thought to spread to the eye [1,2]. VRL is rare (~380 US cases/year) and presents most commonly as diffuse large B cell lymphoma (DLBCL) [1]. VRL can cause vision loss and has a poor prognosis. Radio- and chemotherapy are used empirically and have short-lived responses (progression-free survival (PFS) ~1-year, overall survival (OS) ~3 years). Genomic driver alterations have been described in a limited number of patients by us and others, but precision oncology approaches that use molecularly informed targeted treatments remain in their infancy [3,4].

Diagnostically, vitreous fluid-sampled VRL cells are analyzed by cytology and/or flow cytometry for morphological features and cell surface markers. However, cell scarcity, limited fluid amount, and the gelatinous consistency of the fluid contribute to inaccurate results [5,6]. High-sensitivity assays, such as the detection of the lymphocyte cell receptor V(D)J DNA recombination characteristic of clonal lymphoid populations or the use of next-generation sequencing (NGS) to identify driver genomic alterations, are highly compatible with vitreous liquid biopsy [3] obtained on an outpatient basis or during vitrectomy. However, these molecular assays, especially genomic analysis, are not routinely performed.

While the majority of PCNSL/VRL cases are DLBCLs, an even rarer subset belongs to the T cell subtype. Herein, we report for the first time to our knowledge the molecular diagnostic and genomic analysis of vitreous liquid biopsy samples from a bilateral T cell VRL case in a 63-year-old male. Following a pioneering approach we previously developed for B cell VRL genomic analysis and precision medicine target nomination from vitreous liquid biopsy [3], herein we used a targeted NGS gene panel (Oncomine Cancer Panel [7], one of the assays used in the NCI-MATCH targeted therapy basket trial [8]) to analyze VRL DNA from this patient.

## 2. Methods

### 2.1. Case Selection

The study was conducted with approval from the IRBs of the University of Michigan (UM) and the University of Iowa (UI). We identified a case of T cell VRL in a 63-year-old male who underwent therapeutic vitrectomy. Vitreous samples were genomically analyzed at the UM Kellogg Eye Center and Pathology Department. Clinicopathological information was obtained from the clinical archive.

### 2.2. DNA Analysis

Approximately 3.5 mL of undiluted vitreous fluid from each eye, stored at −80 °C, was centrifuged at 1000× *g* for 5 min. Genomic DNA was extracted from the resulting pellets using the Qiagen AllPrep FFPE DNA/RNA kit (Qiagen, Hilden, Germany), with the following modifications for non-FFPE cells: (1) no de-paraffinization treatment; (2) first 56 °C incubation reduced to 2 min; (3) incubation at 90 °C omitted). DNA samples were quantitated with the Qubit Fluorometer (ThermoFisher, Waltham, MA, USA). T cell receptor (TCR) rearrangement testing was performed with the BIOMED-2 PCR assays (Invivoscribe Technologies, Inc., San Diego, CA, USA) covering ~90% of *TCRB* and *TCRG* rearrangements.

### 2.3. Next-Generation Sequencing

NGS was performed as previously described [3,9]. Briefly, 20 ng of DNA underwent library construction using a targeted custom panel (Oncomine Cancer Panel (OCP), ThermoFisher) targeting 126 cancer-related genes (3435 amplicons). Targets were selected based on large-scale pan-solid tumor and lymphoma genomic data prioritizing recurrent and/or targetable cancer mutations, short insertions/deletions, and copy number alterations (CNAs) [7]. Sequencing and analysis were performed as previously reported [7,10]. Briefly, barcoded libraries were constructed with the AmpliSeq method (Library Kit 2.0, ThermoFisher). Sequencing was performed on IonProton and Torrent Suite 5.0.2. Variant and CNA annotation, filtering, and prioritization were performed as reported using in-house pipelines.

### 2.4. Next-Generation Sequencing Analysis

Data analysis was performed using Torrent Suite 5.0.2, with alignment by TMAP using default parameters, and variant calling was performed using the Torrent Variant Caller plugin (version 4.0-r76860) using default low-stringency somatic variant settings. Called variants were filtered to remove synonymous or non-coding variants, those with flow-corrected read depths (FDP) <20, flow-corrected variant allele containing reads (FAO) <6, variant allele fractions (FAO/FDP) <0.10, extreme skewing of forward/reverse flow-corrected reads calling the variant (FSAF/FSAR <0.2 or >5), or indels within homopolymer runs >4. Called variants were then filtered using a panel-specific, in-house blacklist. Variants with allele frequencies >0.5% in the EXAC database and those reported in EXAC and with observed variant allele fractions between 0.40 and 0.60 or >0.9 were considered germline variants unless occurring at a known hot-spot. Variants located at the last mapped base (or outside) of amplicon target regions, variants with the majority of supporting reads harboring additional mismatches or indels (likely sequencing errors), those in repeat-rich regions (likely mapping artifacts), and variants occurring exclusively in one amplicon if overlapping amplicons cover the variant, were excluded. High-confidence somatic variants passing the above criteria were then visually confirmed in Integrative Genomics Viewer (Broad Institute, Cambridge, MA, USA, https://www.broadinstitute.org/igv/ (accessed on 15 October 2020)). We have previously confirmed that these filtering criteria identify prioritized high-confidence somatic variants that pass Sanger sequencing validation with >95% accuracy [11].

Copy number analysis from total amplicon read counts provided by the Coverage Analysis Plug-in (v4.0-r77897) was performed essentially as described using a validated in-house approach. Log2 copy number ratio was calculated as the amplicon level ratio between read counts in the tumor sample and read counts in a composite of normal samples, normalized for sequencing depth and GC content. Gene-level estimates were calculated as coverage-weighted averages of amplicon-level log2CN ratios. Genes with a log2 copy number ratio estimate of <−1 or >0.6 were considered to have a high level of loss or gain, respectively.

## 3. Results

A 63-year-old male with a past medical history of shingles and anterior uveitis in the left eye six months prior, presented with blurred vision and floaters in both eyes. Other baseline workup reported no relative afferent pupillary defect (diagnostic sign for optic nerve pathology) and normal intraocular pressure (diagnostic sign for pressure-induced vision changes). Most notably, the patient’s vitreous humor contained free-floating cells in both eyes (where there should be none), and the left eye demonstrated whitish deposits in the subretinal space of the central and peripheral retina (Figure 1A–H). PCR of anterior aqueous ocular fluid was negative for herpes simplex virus, cytomegalovirus, or varicella zoster virus infection, ruling out an infectious herpes zoster related etiology. The patient underwent sequential diagnostic and therapeutic vitrectomy, at which time, 3.5 mL of undiluted vitreous was stored for future cytologic and genomic analysis. The patient reported improved symptoms and postoperative visual acuity of perfect vision in both eyes.

Cytology revealed mononuclear cells that were CD20- and CD10-negative and CD3-positive, suggesting vitreous T cell presence (Figure 1I). Genomic DNA (66 ng (OS, oculus sinister, left eye) and 37 ng (OD, oculus dexter, right eye) was extracted from the undiluted vitreous and underwent T cell receptor (TCR) PCR testing. This assay has the sensitivity to detect the 90% of most common V(D)J rearrangements in the *TCRB* and *TCRG* genes [12]. Nearly identical peaks of 186.02 bp (OS) and 186.04 bp (OD) were present at ~5–10% frequency, indicating the presence of clonal T cell populations (Figure 2A). These peaks were present only in the *TCRG* assay.

In order to rule out a non-malignant clonal T cell proliferation, (e.g., infection), NGS using a panel of 126 cancer-related genes, modified from the Oncomine Cancer Panel (OCP) used in the NCI MATCH clinical trial, was performed. Both OS and OD samples showed a scarcity of mutations and short insertions or deletions in the genes assayed. This included a lack of *MYD88* hotspot mutations commonly observed in vitreoretinal and systemic B cell lymphomas. However, copy number alterations (CNAs) were present in nearly identical patterns between the two eyes. These were characterized by a copy gain in the oncogene *BRAF*, an actionable alteration. Genes with copy number loss included the DNA methyltransferase *DNMT3A*, a tumor suppressor and epigenetic DNA-modifying enzyme. Neither sample showed a copy loss in the cell cycle inhibitor *CDKN2A*, a highly recurrent B cell VRL alteration [3]. These CNAs were present at low levels, consistent with the low *TCRG* rearrangement peak predominance of 5–10% (Figure 2B). In order to rule out our observed CNAs being due to sequencing technical variability, we compared the nearly identical OD and OS copy number profiles to a sample obtained from another patient run in the same experiment and initially suspected for VRL but found not to harbor any evidence of cancer (Appendix A, Sample #7). OD/OS copy number profiles differed from Sample #7, thus providing further support for OD and OS containing clonal malignant populations harboring *BRAF* and *DNMT3A* CNAs. Taken together, these data show the feasibility of vitreous liquid biopsy as a source suitable not only for commonly performed morphological and surface marker detection, but also for obtaining precise molecular genomic information in VRL cases. This information supports unambiguous diagnostication and enables nomination of putative precision oncology biomarkers to expand treatment options.

## 4. Discussion

To our knowledge, this work represents the first exploration of the actionable cancer genome of a T cell VRL case. This patient showed a rare clonal T cell population, with no lymphoma seen outside of the eye, and was treated with eight cycles of bilateral intravitreal methotrexate. There was no recurrence and no CNS or systemic lymphoma seen during his disease course. Using NGS, we found a scarcity of mutations/indels but presence of CNAs in an oncogene and tumor suppressor.

Limitations include the “N-of-1” nature of this study which limits generalizability, as well as our limited targeted gene set which does not preclude alterations in genes that were not assayed here. An additional limitation is the low tumor content of both samples (as indicated by the *TCRG* clonal peak dominance of 5–10% and modest CNA levels). Although the evidence suggests presence of clonal malignant T cell populations identical in each eye, a non-malignant cause cannot be definitively excluded based on the available data.

Our data suggest largely distinct genomic landscapes between B cell VRLs and this T cell case in agreement with previous non-ocular, systemic lymphoma studies [3,13,14,15,16]. While *CDKN2A* deleterious alterations (highly recurrent in B cell lymphomas) have been shown to recur in systemic T cell lymphomas [16], our patient had a neutral copy number for this gene. *TP53* alterations, also described in systemic T cell lymphomas [14,16], were absent in this patient. Taken together, our data suggest that T cell VRLs are amenable to genomic analysis for diagnosis and precision oncology from minute vitreous liquid biopsy samples.

## Figures and Tables

**Figure 1 ijms-22-06099-f001:**
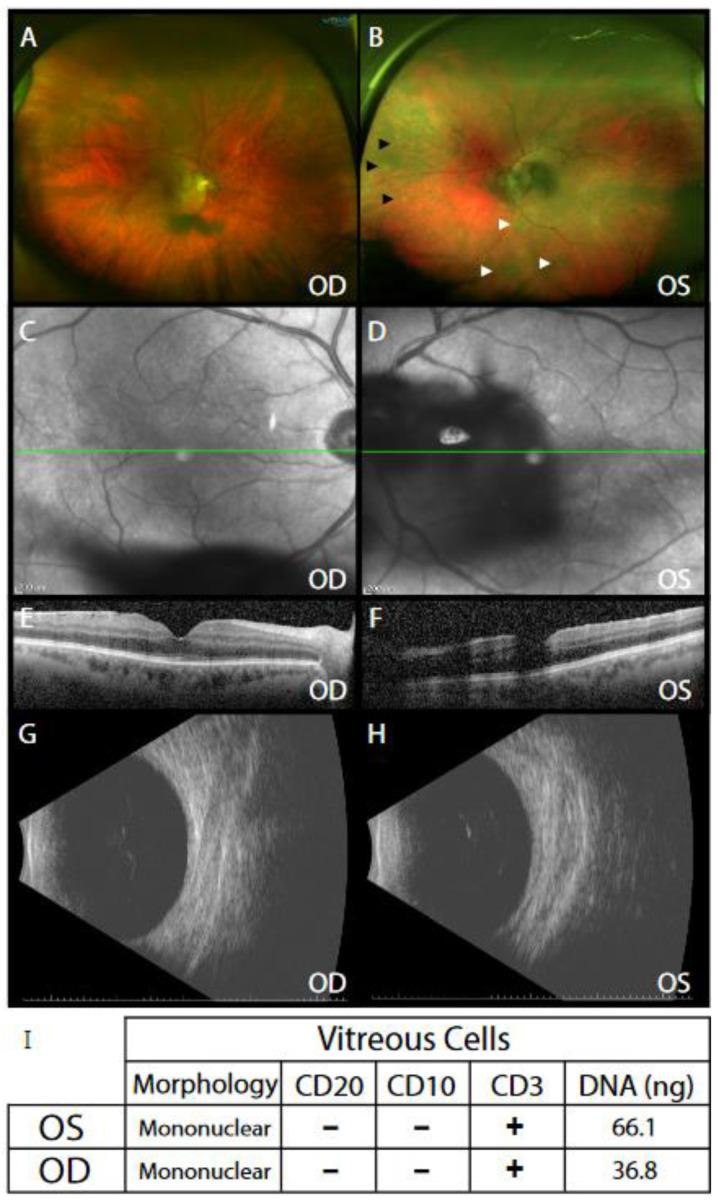
Clinical examination shows vitreous debris in both eyes. (**A**) Ultra-widefield retinal imaging of the right eye (OD) showed vitreous clumps over the posterior pole. (**B**) Ultra-widefield retinal imaging of the left eye (OS) showed vitreous cells (black arrowheads); inferior snowballs (white arrowheads); and dense vitreous clumps over the posterior pole. (**C**,**D**) Infrared en-face imaging shows level (green-line) at which optical coherence tomography (OCT) was used to image cross-section of macula (**E**,**F**). Cross-sectional OCT showed (**C**) mild epiretinal membrane OD and (**D**) a dense vitreous opacity over the macula OS. Posterior B-Scan ultrasonography revealed vitreous opacities (**G**) OD and (**H**) OS. (**I**) Cytological examination for morphology and cell-surface marker staining for CD20, CD10 and CD3 as well as vitreous DNA amounts are shown for each vitreous sample.

**Figure 2 ijms-22-06099-f002:**
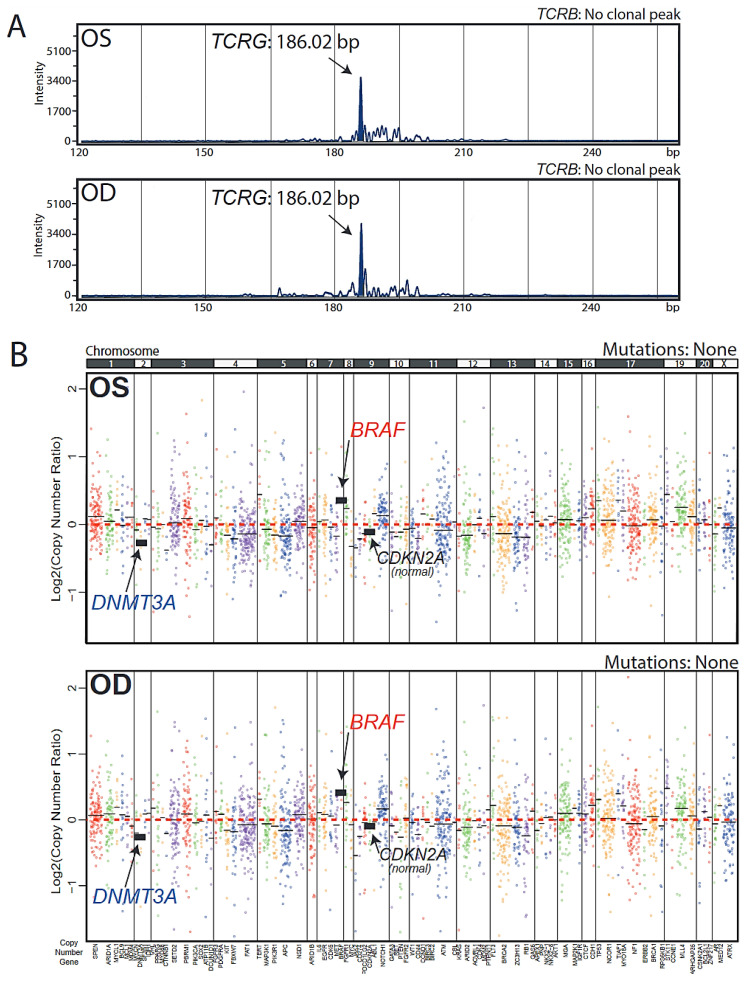
DNA analysis of vitreous liquid biopsy samples. (**A**) DNA obtained from vitreous fluid from each eye underwent clonal T cell receptor (TCR) rearrangement PCR testing covering ~90% of the most common rearrangements in the *TCRB* and *TCRG* genes. Assays were run in duplicate with negative controls (one replicate shown per sample). Clonal rearrangements were defined as peaks identical between replicates that were at least 2× higher than the third highest peak. Capillary electropherogram plots of band intensity over fragment length show a positive PCR result for *TCRG* rearrangement at 186 bp, identical in both eyes, at ~10% frequency. No TCRB-positive bands were observed in either sample. (**B**) NGS using the OCP version 1 panel was performed on each eye sample. Copy number plots are shown as log2 copy number ratio (amplicon level ratio between read counts in the tumor sample and read counts in a composite of normal samples, normalized for sequencing depth and GC content). Dots represent individual amplicons, dots of the same color represent a gene, and black horizontal bars represent average gene-level estimates (coverage-weighted). Altered/relevant genes are highlighted. No mutations were observed in either sample.

## Data Availability

Data reported in this study are available upon request to author Rajesh C. Rao.

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
