# Peer review of "Molecular Characterization of a Rare Case of Bilateral Vitreoretinal T Cell Lymphoma through Vitreous Liquid Biopsy"

_ijms, 2021, doi:10.3390/ijms22116099_

Round 1
Reviewer 1 Report
In the present study, the authors report the molecular diagnostic and genomic analysis of vitreous liquid biopsy samples from a bilateral T cell VRL case in a 63-year-old male. To their knowledge, this work represents the first exploration of the actionable cancer genome of a T cell VRL case.
The molecular diagnostic workup of this rare case of bilateral T cell VRL is reported as well as the characterization of its genomic landscape including identification of potential targetable alterations using next generation sequencing.
While CDKN2A deleterious alterations are highly recurrent in B cell lymphomas, and also having been shown to recur in systemic T cell lymphomas, their patient had a neutral copy number for this gene. Also, TP53 alterations, described in systemic T cell lymphomas, were absent in this patient.
However, they found that copy number alterations (CNAs) were present, in nearly identical patterns between the two eyes. These were characterized by copy gain in the oncogene BRAF, an actionable alteration. Also, genes with copy number loss included the DNA methyltransferase DNMT3A, a tumour suppressor.
Thus, their data suggest largely distinct genomic landscapes between B cell VRLs and this T cell case in agreement with previous non-ocular, systemic lymphoma studies.
The authors claim that taken together their data suggests that T cell VRLs are amenable to genomic analysis for diagnosis and precision oncology from minute vitreous liquid biopsy samples.
Conclusion:
The novelty of this paper is unquestionable. The limitation is though the “N-of-1” nature of the study which limits generalizability, as well as the limited targeted gene set which does not preclude alterations in genes that were not assayed, as well as the low tumour content of both samples.
However, they used next generation sequencing (NGS) to identify driver genomic alterations, targeting 126 cancer-related genes (3,435 amplicons), using a targeted custom panel (Oncomine Cancer Panel, one of the assays used in the NCI-MATCH targeted therapy basket trial) to analyze VRL DNA from this patient.
The limitations of the study are clearly acknowledged by the authors. Saying that, one should also acknowledge that vitreoretinal lymphoma (VRL) is an uncommon eye malignancy, and especially VRLs of T cell origin are rare. Therefore, a single case thorough study like this should be commended.
The authors conclude “that these data show the feasibility of vitreous liquid biopsy as a source not only suitable for commonly performed morphological and surface marker detection, but also for obtaining precise molecular genomic information in VRL cases. This information supports unambiguous diagnostication as well as enables nomination of putative precision oncology biomarkers to expand treatment options”, emphasizing that the findings being highly compatible with vitreous liquid biopsies obtained on an outpatient basis or during vitrectomy.
To this reviewer the results are interpreted appropriately, the results being of scientific and clinical significance. The conclusions are justified and supported by the results. In addition, the results are presented systematically.
Furthermore, the study is well designed and technically sound. Analyses of data are performed with highest technical standards. Data are robust, sufficient for the conclusions drawn.
This paper should be of interest to medical ophthalmologists, oncologists, radiotherapists and molecular biologists with interest in translational research.
Thus, this study definitely represents an advance in precision oncology for the treatment of vitreoretinal T cell lymphoma through vitreous liquid biopsy.
Finally, just a few remarks:
Shouldn’t in Fig.1, description of A in the figure be B ..?
In line 169: “…DNMT3A, tumor suppressor” > DNMT3A, a tumor suppressor.
Author Response
Response: We thank the reviewer for the careful consideration of our manuscript and for correctly pointing out its strengths as well as its limitations. We continue to accumulate any ultra-rare vitreoretinal T cell (and B cell) lymphoma samples that we can locate. Over time we will hopefully assemble a sizeable cohort that will allow us to generate knowledge on the tumor biology and clinical significance of this disease.
“Shouldn’t in Fig.1, description of A in the figure be B ..?”
We thank the reviewer for pointing this detail out. Yes, the reviewer is absolutely correct, the right eye and left eye labeling in the figure legend was reversed. This has been corrected. We regret this mistake.
“In line 169: “…DNMT3A, tumor suppressor” > DNMT3A, a tumor suppressor.”
We thank the reviewer for this careful and correct observation. We have added the “a” as suggested.
Reviewer 2 Report
This paper reports a rare case of bilateral vitreoretinal T cell lymphoma with no other lymphoma lesions outside of the eye, and the patient was successfully treated with intravitreal methotrexate alone. The major concern is lack of convincing evidences supporting that the diagnosis is accurate, and the copy number alterations of BRAF and DNMT3A alone seem to be weak to underpin the pathophysiology of this rare case.
Author Response
Reviewer Comments: This paper reports a rare case of bilateral vitreoretinal T cell lymphoma with no other lymphoma lesions outside of the eye, and the patient was successfully treated with intravitreal methotrexate alone. The major concern is lack of convincing evidences supporting that the diagnosis is accurate, and the copy number alterations of BRAF and DNMT3A alone seem to be weak to underpin the pathophysiology of this rare case.
Response: We appreciate the concern presented by the reviewer and we agree that a non-malignant etiology cannot be definitively and irrefutably be excluded, as we mention in the discussion. With respect to the diagnosis, the patient was cared for under the official diagnosis of a vitreoretinal lymphoma, which included a combined diagnostic and therapeutic vitrectomy, as well as close follow up. Additional testing for other causes of inflammatory and infectious etiologies were explored and detailed in the results of the manuscript.
With respect to the conclusion of this being a T cell lymphoma, we performed capillary electropherography of PCR testing for T cell and B cell receptor rearrangement detection, both of which confirmed the cells were in fact clonal T cells. We believe we performed this testing to the strictest standards to be sure we didn’t overlook any disparities in the data. As shown in the table in Figure 1I, cytology revealed a CD20 (B-cell) negative population that was almost exclusively CD3 (T cell) positive, supporting a non-infectious cause.
With respect to the copy number alterations, we note that we used a commonly-used 126-oncogene panel. Given the number of genes that could possibly drive this cancer, we concede that there is always a small chance the 126-gene panel may not have the driving mutation. However, we believe our analysis of the copy number gain and loss used very conservative metrics in order to guarantee that any real change was true and not noise in the data. Further, the fact that those copy changes are not prominent is consistent with the relatively low tumor fraction of the samples from T cell receptor rearrangement PCR. As we have previously reported (reference #10), low tumor content compresses copy number plots toward the normal diploid line. The presence of copy-number changes identical between eyes as well as identical T cell receptor rearrangements means that although not clear-cut as the reviewer mentions, the evidence weighs on this case being a T cell lymphoma. The larger point we aim to make in this work is precisely illustrated by the reviewer’s point about this rather difficult case: current clinical tools often limit a definitive diagnosis. However, inclusion of novel molecular approaches can help improve specificity and sensitivity in order to enable more accurate diagnosis and nominate precision oncology or immuno-oncology therapeutic opportunities.
Round 2
Reviewer 2 Report
This paper reports a rare case of bilateral vitreoretinal T cell lymphoma with no other lymphoma lesions outside of the eye, and the patient was successfully treated with intravitreal methotrexate alone. Although the accurate diagnosis in such a case is often difficult, this paper may provide a helpful approach in such a case.